# OneChart: Purify the Chart Structural Extraction via One Auxiliary Token

## ABSTRACT

Chart parsing poses a significant challenge due to the diversity of styles, values, texts, and so forth. Even advanced large vision-language models (LVLMs) with billions of parameters struggle to handle such tasks satisfactorily. To address this, we propose OneChart: a reliable agent specifically devised for the structural extraction of chart information. Similar to popular LVLMs, OneChart incorporates an autoregressive main body. Uniquely, to enhance the reliability of the numerical parts of the output, we introduce an *auxiliary token* placed at the beginning of the total tokens along with an additional decoder. The numerically optimized (auxiliary) token allows subsequent tokens for chart parsing to capture enhanced numerical features through causal attention. Furthermore, with the aid of the auxiliary token, we have devised a self-evaluation mechanism that enables the model to gauge the reliability of its chart parsing results by providing confidence scores for the generated content. Compared to current state-of-the-art (SOTA) chart parsing models, *e.g., DePlot, ChartVLM, ChartAst*, OneChart significantly outperforms in Average Precision (AP) for chart structural extraction across multiple public benchmarks, despite enjoying only 0.2 billion parameters. Moreover, as a chart parsing agent, it also brings 10%+ accuracy gains for the popular LVLM (LLaVA-1.6) in the downstream ChartQA benchmark.

## CCS CONCEPTS

• **Computing methodologies** → **Computer vision**; *Natural language generation*.

## KEYWORDS

Chart structural extraction, Vision-language model, Multi-modal large language models

## 1 INTRODUCTION

Charts and plots, as key visual language, permeate every aspect of education and work. They help people easily and accurately understand, compare, and analyze data. Beyond just titles, axes, and legends, charts are made up of points, lines, angles, colors, and shapes. These detailed visual elements greatly increase the complexity of automatically parsing charts, making it a challenging yet essential area of research in computer vision [1, 2].

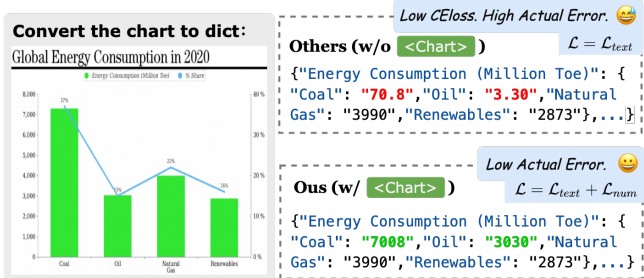

**Figure 1: Schematic diagram comparing our method with other methods. `<Chart>` is the auxiliary special token. The numbers highlighted in red and green represent incorrect and correct predictions, respectively.**

Previous methods [3–6] rely on traditional techniques like detection [7] and Optical Character Recognition (OCR) to transform images into tables, then fine-tuned specialized TableQA models [8, 9] for inference. It is reasonable that comprehensive and accurate perception can effectively assist in information extraction and downstream reasoning tasks. In recent years, with the evolution of vision-language models (VLMs) [10–18], end-to-end chart understanding models such as MatChart [19], ChartAst [20], and ChartVLM [21] started to surface. These models meld vision encoders and autoregressive decoders, aiming for pre-training on image-to-table tasks and fine-tuning for Question and Answer (QA) applications. Despite their advances, accoring to our experiments in Section 4.3, these models with billions of parameters still face limitations in extracting structured information and processing various chart styles, especially in the scenario of parsing charts lacking numerical annotations.

We think the performance issue seen in the above VLMs is primarily due to two factors. Firstly, the vision encoder may exhibit the issue of "CLIP bias". Most of the models mentioned employ a CLIP-based [22] ViT as the vision encoder. However, since CLIP-ViT is primarily trained on short, global descriptions of natural image-caption pairs, using it as a vision encoder may lead to the omission of crucial local details necessary for chart parsing. This discrepancy could result in a gap between CLIP-ViT's functionality and tasks that require dense perception (such as chart parsing). Additionally, the training mainly conducted with English captions also affects the effectiveness of the CLIP-ViT in encoding charts embedded in other languages. Secondly, the use of cross-entropy loss in autoregressive decoders presents limitations in accurately capturing or predicting numerical values. For instance, the cross-entropy losses for the numbers "7008" and "70.8" shown in Figure 1 can be deceptively similar. This proximity in loss values complicates the model's convergence process and reduces its accuracy in capturing numerical values in charts. Moreover, there are limited

diversified public benchmarks in chart parsing filed. ChartQA [3] and PlotQA [4] primarily consist of bar and line charts from online platforms, with very few pie charts included. Similarly, datasets like SimChart9K [23] and ChartX [21], created using Matplotlib, show limited stylistic variety. The lack of diverse benchmarks for chart analysis hinders the development of related research areas.

To tackle these challenges, we introduce a chart converter named OneChart in this work. It captures essential components like chart titles, sources, and aligned numerical data and maps them to a Python-dict format, which can effectively facilitate downstream chart reasoning tasks. To overcome the "CLIP bias" mentioned above and enhance the model's ability to compress chart information, we train a specialized chart encoder from scratch using a large amount of synthetic chart data in both English and Chinese. In parallel, to elevate the model's capability to interpret numerical values in charts and increase the reliability of its numerical output, an additional auxiliary token is introduced. We also develop a decoder specifically for this token and optimize it using a customized L1 loss. Moreover, we present a large-scale chart-to-dict benchmark named ChartY, which comprises approximately 6K charts. These charts span a broad spectrum of topics and types and include content in both English and Chinese languages. In sum, our primary contributions are as follows:

- **Introduction of OneChart:** We propose OneChart, a state-of-the-art chart-to-dict model that uses an auxiliary token to guide the model towards more accurate numerical value parsing. This model serves as a foundational framework for other researchers to further develop and enhance.
- **Creation of the ChartY benchmark:** We standardize the tasks involved in chart-to-dict and introduce a new, comprehensive benchmark ChartY. This benchmark spans a wide array of topics, chart types, and languages, offering a robust platform for future research and evaluation.
- **Numerous experiments and analyses:** Experiments reveal that OneChart achieves SOTA performance in structural extraction. It shows a 19.1% to 29.4% improvement compared to suboptimal methods particularly in charts lacking numerical annotations. Additionally, the integration with popular VLMs enhances accuracy by 32.6% with LLaVA1.5 [11] and 11.2% with LLaVA1.6 [24] on the ChartQA benchmark.

## 2 RELATED WORK

### 2.1 Chart Structural Extraction

Chart structural extraction aims to extract the main textual and visual elements (such as title, axis names, legends, values, etc.) from chart images through certain methods or models, and organize them in an appropriate way. In the early stage, some non-end-to-end methods used keypoint/region detection or segmentation methods, combined with OCR and other methods for information extraction [3, 4, 6, 25–27]. While these methods have advanced the extraction and analysis of chart structures, their implementation is complex and heavily reliant on the generalization capabilities of traditional techniques. These methods are commonly used for specific types of tables and have lower generalization performance for real-world charts. Currently, several studies [19, 20, 23, 28–30] tend to use the vision-language models (VLMs) to extract the

information contained in visual charts end-to-end and store it in a table format. This approach effectively translates visual data into linguistic formats. Beyond just transforming chart data into tables, ChartVLM [21] also decouples the task of parsing chart titles.

### 2.2 Chart Reasoning

Chart reasoning aims to provide relevant descriptions, summaries, QA, or comparative analysis of visual charts. At present, researches are mainly divided into two-stage and end-to-end methods, which treat chart reasoning as a downstream task after extracting key information from charts. PlotQA [4] and ChartQA [3] extract the key information and send it to TableQA models [8, 9, 31] for reasoning and answering. StructChart [23] and DePlot [28] utilize the inference ability of pre-trained large language model [32, 33] with a small number of shots, and use the output as the prompt for reasoning. End-to-end approaches like ChartAssistant [20], ChartLlama [30], and ChartVLM [21] start by aligning visual charts with their textual information through pre-training from charts to tables. They then fine-tune various tasks including information extraction, open question answering, and summarization, enabling simultaneous implementation of information extraction and downstream tasks. Clearly, whether using end-to-end or two-stage methods, the structural extraction of information from charts remains fundamental.

### 2.3 Multimodal Chart Benchmarks

At present, there is not a lot of open source benchmarking work. ChartQA [3] and PlotQA [4] are mainly suitable for tasks such as chart-to-table and QA summary. Chart-to-Text [34] is mainly suitable for chart-to-table and summary tasks, but the truth quality of the table is poor. The current benchmark works such as StructChart [23], MMC [35], and ChartVLM [21] cover more tasks, such as code redrawing, analysis, and type judgment. These works have to some extent promoted the development of chart parsing work. However, the data used to evaluate multimodal large-scale models for charts is still relatively limited in terms of style, type, and language diversity.

## 3 METHOD

In this section, we outline the methodology behind OneChart, structured into five key areas: Data Engine, Architecture, The Auxiliary Token, Training Process, and Inference. Each part plays a critical role, from providing training data and defining structural design to detailing our approach, optimization strategies, and inference results.

### 3.1 Data Engine

**Chart data generation**. Except for chart data from online platforms, such as ChartQA, most chart data is generated using tools such as Matplotlib and Pyecharts. Consequently, we utilize both tools to generate chart images. The charts generated by Matplotlib all contain four fields: "title", "x_axis", "y_axis" and "chart body". Due to the limited functionality of Matplotlib and Pyecharts, we specifically introduce the "chart source" to better fit the real-world chart data style. In addition to taking general rendering methods,

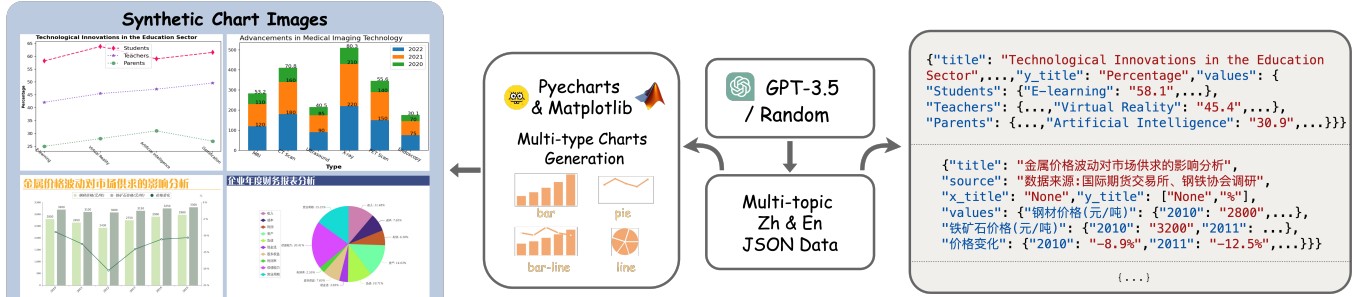

**Figure 2: Process of our data generation. We randomly generate multi-topic source data in both Chinese and English using GPT-3.5 or random corpora. Subsequently, we employ two rendering tools, Matplotlib and Pyecharts, to produce chart of various styles and types.**

we add an additional two-stage rendering method, which first creates the main part except for the title and source, and then adds the title and source to the chart stochastically through graphic stitching. To enhance the visual diversity of charts, we employ randomly generated 16-bit color codes to alter the colors of both text and graphics, beyond the commonly used color schemes. We also offer hundreds of distinct text fonts. Additionally, we introduce considerable variability in the size, direction, and quantity of visual elements. For the generation of pre-training data, the content of the charts is produced randomly. Specifically, for textual information such as title and source, we utilize the Natural Language Processing (NLP) corpus [36], extracting entries randomly by setting predetermined lengths. The numerical content is generated under controlled distribution to ensure variability. In total, the process yields about 10M chart images alongside their corresponding truth labels. Figure 2 shows the process of our data generation.

**Data details**. The data we generate predominantly fall into two principal categories: barline and pie charts. **(1) Barline Charts**: These are categorized into five distinct types: Single Column Chart, Multi Column Chart, Single Line Chart, Multi Line Chart, and Combo Chart (Mixed Chart). Each type is evenly split between visualizations that feature numerical labels and those that do not. Currently, our Barline charts can accommodate up to three legends. **(2) Pie Charts**: In this category, Labeled Pie Charts and Pie Charts with Legends are distributed in equal proportions. Furthermore, in the process of generating content with logical and practical significance using GPT-3.5, we employ varied prompts to facilitate the creation of thematically diverse data across several domains, such as finance, education, technology, among others.

## 3.2 Architecture

OneChart is an end-to-end chart information extraction tool based popular VLM architecture, as shown in Figure 3. Regarding the selection of VLMs, we choose for the recently released Vary-tiny model [15], which consists of a vision encoder from SAM-base and a tiny auto-regressive OPT-125M [37] decoder, linked by a linear layer to synchronize their channel dimensions.

For the chart image input, we simply resize the image to a fixed resolution 1024×1024 without any extra data augmentation. The model learn to extract Python-dict information with respect to the input image using causal masked language modeling, which can be written as:

$$\mathcal{L}_{text}(\theta, w) = -E_{(w,v)\sim D} \log P_\theta (w_m \mid w_{<m}, v) \quad (1)$$

where $w$ denotes the target text sequence, $v$ denotes the vision features from the vision backbone, $m$ denotes the current index of the output target token and $D$ denotes the dataset.

## 3.3 The Auxiliary Token

To enhance the reliability of number values in outputs and mitigate the risk of significant errors, we introduce a special token denoted as "<Chart>" by prefixing its output. This special token will trigger an extra chart's numerical values prediction. As shown in Figure 3, the corresponding hidden state embedding $t \in \mathcal{R}^{768}$ of auxiliary token "<Chart>" is fed into a auxiliary decoder $\mathcal{F}$ comprised of 3 layer MLPs and 2 ReLU activation function. The auxiliary output denoted as $\mathcal{F}(t)$, where $\mathcal{F}(t) \in \mathcal{R}^{256}$ represents the normalized numerical values prediction within a chart. To supervise the numerical output, we incorporate the L1 loss for the number loss $\mathcal{L}_{num}$ during training:

$$\mathcal{L}_{num}(\theta, u) = E_{(u,t)\sim D}|\mathcal{F}(t) - u|_{masked}, \quad (2)$$

where $u$ represents the min-max normalized ground truth values within a chart image. Each vector of ground truth values is extended to a fixed length of 256 elements through padding with "nan" values to facilitate parallelized training across batches. In the loss function $\mathcal{L}_{num}$, $masked$ is the non-padded (non-nan) elements, ensuring that the padding does not influence the loss computation.

Since OPT-125M in OneChart is a transformer-based model incorporating causal attention, it can attend to the hidden state of the first <Chart> entries when processing the text output in the form of a Python-dict. The auxiliary decoder is an optional component rather than a primary tool during inference. This design maintains the model's versatility and ease of use, akin to the traditional vision-language model (VLM). Additionally, the auxiliary number decoder can participate in the computation of confidence scores to help filter its predictions, thereby enhancing the output's reliability. A detailed exploration of this process is presented in Section 3.5.

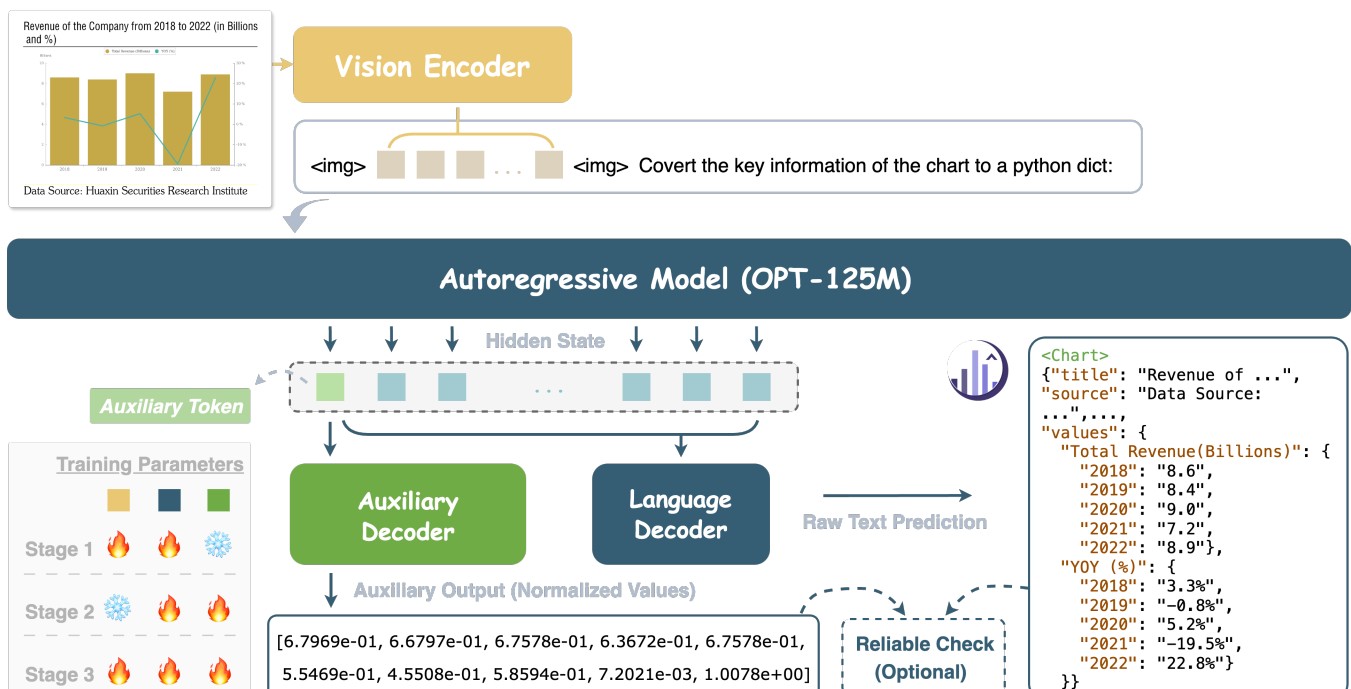

Figure 3: Overall pipeline of OneChart model. Compared with popular VLMs, we introduce an additional auxiliary token `<Chart>` at the start of the token sequence, alongside an extra decoder, to enhance the reliability of the numerical outputs.

## 3.4 Training Process

Along all the training stage, we use the template of Vicuna v1 [38] to construct ground truth in a conversation format as "USER: `` `[image]` `</img>` Covert the key information of the chart to a Python-dict. ASSITANT: `<Chart>` `[texts output]` ``". We add the "``", "`<Chart>`" and "`</img>`" as special tokens of the text tokenizer of OPT-125M and we find that it can adapt very well to the Vicuna template. `[image]` represents the vision feature that occupies 256 tokens and `[texts output]` is the Python-dict format text of the chart.

**Stage1: Pretraining.** During this stage, we perform pre-training using 10 million synthetic chart data, including Chinese and English languages. The chart of 5 million is generated by matplotlib, and the other 5 million is generated by pyecharts. The data source of chart is randomly generated in this stage. The model is trained with a batch size of 16 and a learning rate of 1e-4 for 3 epochs. During this stage, the entire vision encoder with language model are trained. The training loss is formulated as:

$$\mathcal{L}_{stage1} = \mathcal{L}_{text} \tag{3}$$

where $\mathcal{L}_{text}$ is cross entropy loss. The Stage1 training uses 32 A100 (80G) GPUs for around 12 hours.

**Stage2: Warmup auxiliary number decoder.** In the second stage, we use about 2.7 millions SFT data as shown in Table 1 to warmup auxiliary number decoder. In this stage, we frozen the vision encoder and only train language model and auxiliary decoder.

The training loss is defined as:

$$\mathcal{L}_{stage2} = \mathcal{L}_{text} + \mathcal{L}_{num} \tag{4}$$

In Stage2, we use batch size of 16 and a learning rate of 5e-5 for 1 epoch, this training uses 16 A100(80G) GPUs for around 3 hours.

**Stage3: Supervised Fine-tuning (SFT).** In this stage, we fine-tuning total model parameters utilizing above SFT data. The training loss $\mathcal{L}_{stage3}$ is same as $\mathcal{L}_{stage2}$. We use batch size of 16 and a learning rate of 5e-5 for 1 epoch, this training uses 24 A100(80G) GPUs for around 4 hours. Subsequently, we employed this fine-tuned model to evaluate its performance across all benchmarks in Section 4, recording the scores achieved.

## 3.5 Inference

During inference, the provided chart image is first resized to 1024 × 1024 pixels, with the pixel values scaled to the range of 0 to 1. Subsequently, through a vision encoder, a vision embedding $v \in \mathcal{R}^{256 \times 768}$ is integrated into the text embedding of the Vicuna v1 conversation template, as described in Section 3.4. The text in Python-dict format is then serialized for output. The appearance of the "``" special token signifies the end of the output, which we refer to as the raw output. The performance of this raw output is presented in the SE benchmark in Table 2 and the OCR benchmark in Table 2.

Moreover, and critically, we introduce an **option** to incorporate the output from the auxiliary number decoder to assess the reliability of the raw output. As illustrated in Figure 4, the raw predict can be easily parsed into a dictionary in python by `json.loads()`

**Table 1: Overview of fine-tuning data sources and samples in English (En.) and Chinese (Zh.) Some charts come from real-world online platforms (Real.) and others are rendered by Python. ChartQA: images from the ChartQA training set, PlotQA: images from the PlotQA training set.**

| Lang. | Data | Source | Render | Samples |
|---|---|---|---|---|
| En. | ChartQA. | Real. | – | 17.8 K |
| | PlotQA. | Real. | matplotlib | 157 K |
| | pye-barline. | GPT3.5 | pyecharts | 640 K |
| | pye-pie | GPT3.5 | pyecharts | 184 K |
| | reversal. | GPT3.5 | pyecharts | 50 K |
| | mat-barline. | GPT3.5 | matplotlib | 640 K |
| | mat-pie | GPT3.5 | matplotlib | 184 K |
| Zh. | pye-barline. | GPT3.5 | pyecharts | 640 K |
| | pye-pie | GPT3.5 | pyecharts | 184 K |
| Total | | | | 2.7 M |

function. Following this, the numbers are extracted from the dictionary in sequence and subjected to min-max normalization, denoted as $u_r$. Simultaneously, the auxiliary decoder generates a numerical prediction, $u_c$. The self-consistency distance $\mathcal{S}$ between these two types of predictions is calculated as follows:

$$\mathcal{S} = \frac{1}{N} \sum_{i=1}^{N} |u_{r\,i} - u_{c\,i}| \tag{5}$$

where $N$ represents the number of numeric values contained within the "value" field of the raw output. The range of $\mathcal{S}$ is between $[0, 1]$. The smaller the value, the closer the self-consistency distance, which can also be explained as the model being more confident in its own output accuracy. Additionally, by setting a threshold, we can quantify the "quality" of the output results, guiding users to selectively trust the outputs.

## 4 EXPERIMENTS

### 4.1 Evaluation Metrics

Model should extract the key elements of chart in the defined Python-dict structure. Predicted dictionary should at least include these elements: "title", "source", "x_axis", "y_axis", and "value". We comprehensively evaluate the model's output from two aspects: textual OCR accuracy and structural extraction precision. We also report the accuracy in the popular QA benchmark.

**Textual OCR**. For the chart's textual elements such as "title", "source", "x_axis", "y_axis" in dictionary, we employ an accuracy evaluation based on normalized edit distance [39, 40], which allows us to measure the closeness of the model-generated text to the ground truth with precision. In order to unify the evaluation indicators as larger is better, we report the value of 1 minus the normalized editing distance as the OCR accuracy, denoted as Reverse Edit distance (RE).

**Structural Extraction**. For the "values" field, which is itself also a Python-dict, represents the entity name and numerical data presented in the chart. To evaluate the accuracy of this crucial

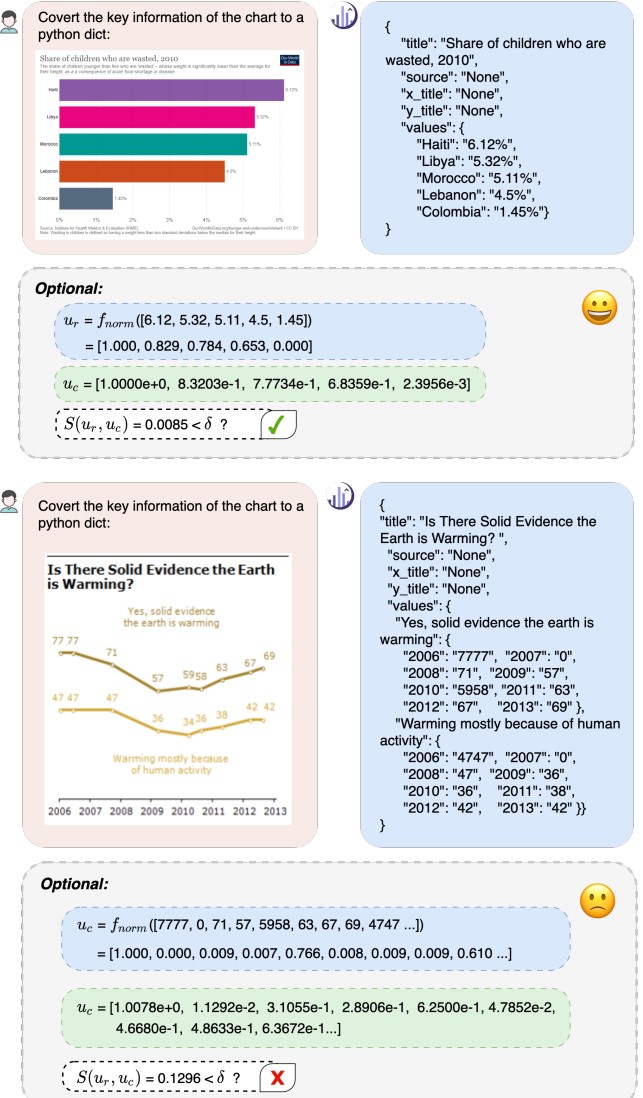

**Figure 4: Inference pipeline. OneChart directly outputs raw results in text and has an optional self-consistency distance to determine the reliability of the raw predicts.**

component, we concatenate the key and item pairs into tuples and assess them using the mean Average Precision (AP) from the SCRM (Structuring Chart-oriented Representation Metric) [23]. According to the definition of SCRM, three different levels of tolerance ($tol := \{strict, slight, high\}$) are set for fine-grained evaluation of SE task:

$$
\begin{aligned}
strict &:= \left\{ J_{thr}|_{tol} = 0 \land e_{thr}|_{tol} = 0 \right\}, \\
slight &:= \left\{ J_{thr}|_{tol} = 2 \land e_{thr}|_{tol} = 0.05 \right\}, \\
high &:= \left\{ J_{thr}|_{tol} = 5 \land e_{thr}|_{tol} = 0.1 \right\},
\end{aligned} \tag{6}
$$

where $J_{thr}|_{tol}$ indicates the edit distance threshold between prediction and GT string, $e_{thr}|_{tol}$ refers to the relative error threshold between prediction numeric value and GT value.

**Table 2: Average precision (AP) is evaluated using SCRM for the Structural Extraction (SE) task, and Title, Source, Y_axis OCR are evaluated using Reverse Edit distance (RE). ChartVLM refers to ChartVLM-Base-7.3B. AP 🟥 represents AP@strict, AP 🟨 represents AP@slight, and AP 🟩 represents AP@high. The best result is shown in Bold, and the second-best result is underlined. The results of OneChart are highlighted in ▒ light blue ▒.**

| Method | Size | Metric | ChartQA-SE | PlotQA-SE | ChartX-SE | | | | | ChartY-en | | ChartY-zh | |
|---|---|---|---|---|---|---|---|---|---|---|---|---|---|
| | | | | | bar | barnum | line | linenum | pie | barline | pie | barline | pie |
| **Numerical Values Marked** | | | Partial | No | No | Yes | No | Yes | Yes | Partial | Partial | Partial | Partial |
| ***Structural Extraction*** | | | | | | | | | | | | | |
| DePlot [28] | 1.3B | AP 🟥 | 61.41 | 3.11 | 2.20 | 33.70 | 16.00 | 22.30 | 0.00 | 13.16 | 0.05 | 3.33 | 0.00 |
| | | AP 🟨 | 70.89 | 16.49 | 21.70 | 41.30 | 51.20 | 52.90 | 0.00 | 23.78 | 0.88 | 6.28 | 0.00 |
| | | AP 🟩 | 72.88 | 26.50 | 42.10 | **48.70** | 60.10 | **61.20** | 0.00 | 32.88 | 2.29 | 16.14 | 0.02 |
| ChartVLM [21] | 7.3B | AP 🟥 | 71.84 | 3.81 | 10.60 | 20.40 | 26.30 | 29.10 | 40.70 | 15.71 | 6.88 | 3.80 | 0.00 |
| | | AP 🟨 | 81.35 | 46.83 | 17.70 | 27.50 | 42.90 | 45.00 | 41.50 | 29.49 | 7.91 | 5.75 | 0.34 |
| | | AP 🟩 | 84.20 | 54.00 | 21.20 | 33.00 | 51.90 | 54.80 | 43.20 | 38.48 | 11.04 | 15.88 | 7.74 |
| ChartAst [20] | 13B | AP 🟥 | 39.67 | 5.18 | 7.80 | 22.10 | 8.20 | 11.50 | 44.30 | 5.27 | 8.98 | 0.07 | 0.00 |
| | | AP 🟨 | 67.91 | 48.67 | 21.70 | 33.80 | 40.10 | 35.20 | 53.00 | 13.84 | 11.28 | 0.27 | 0.08 |
| | | AP 🟩 | 73.27 | 56.08 | 38.40 | 44.60 | 48.00 | 41.70 | 63.70 | 16.23 | 15.00 | 0.41 | 2.28 |
| Ours | 0.2B | AP 🟥 | **72.02** | **34.56** | **29.70** | **37.22** | **49.30** | **43.18** | **63.33** | **68.43** | **74.95** | **83.19** | **63.80** |
| | | AP 🟨 | **82.91** | **84.18** | **39.45** | **42.50** | **59.79** | **56.55** | **67.33** | **83.13** | **79.07** | **92.49** | **74.96** |
| | | AP 🟩 | **85.94** | **86.10** | **47.92** | 48.14 | **65.25** | 61.19 | **76.10** | **86.76** | **84.32** | **94.65** | **83.82** |
| ***Title OCR*** | | | | | | | | | | | | | |
| ChartVLM | 7.3B | Title-RE | 79.26 | 99.49 | **97.70** | **97.82** | **96.94** | **97.03** | **95.62** | 97.87 | **99.07** | 14.44 | 24.07 |
| Ours | 0.2B | Title-RE | **97.80** | **99.94** | 96.68 | 96.74 | 96.54 | 95.70 | 93.24 | **99.45** | 98.66 | **98.97** | **99.96** |
| ***Other Textual OCR*** | | | | | | | | | | | | | |
| Ours | 0.2B | Source-RE | – | – | – | – | – | – | – | 72.50 | 72.88 | 99.21 | 99.91 |
| | | Y_axis-RE | – | – | – | – | – | – | – | 90.31 | – | 98.31 | – |

**QA**. With OneChart's support, we evaluate some LLM and VLMs on chart-related QA tasks like ChartQA[3]. In this downstream task, we use their vanilla metrics for a fair comparison with other methods.

## 4.2 Benchmark for Structural Extraction

Traditional QA benchmarks, such as ChartQA and PlotQA, often limit their scope to querying small, isolated segments of information from charts, such as individual values, which may not effectively gauge a model's ability to extract and understand the full spectrum of data presented in a chart. In contrast, we aim to establish a benchmark centered around the Structural Extraction (SE) task, which directly assesses the model's accuracy in converting chart images into structured Python-dict representations.

Our benchmark including several parts, which are meticulously composed of images from the following sources, each contributing uniquely to the breadth of the dataset:

- ChartQA-SE and PlotQA-SE: The images for these components are derived from the test sets of ChartQA and PlotQA, respectively. Both datasets originate from real-world online platforms, encompassing a wide range of topics such as economy, finance, society, politics, and industry. Most images in ChartQA have specific numerical annotations on them. PlotQA features charts rendered by software based

on real-world data, including horizontal bar plots, vertical bar plots, line plots, and scatter plots. There are no specific numerical annotations on the images. This part of the benchmark aims to assess the model's effectiveness in recognizing and extracting information from charts that mirror real-life complexities.

- ChartX-SE: This benchmark is derived from the ChartX test set and includes bar, line, and pie charts rendered using Matplotlib. Some charts contain numerical annotations ("barnum", "linenum" and "pie"), while others do not ("bar" and "line"). The selection of ChartX-SE is instrumental in evaluating the model's ability to process and understand conventional chart formats that are prevalent in academic settings.

- ChartY-en and ChartY-zh: Recognizing the diversity limitation in existing datasets, we have augmented benchmark with additional pyecharts rendered images, including both English and Chinese languages. Only some images contain numerical annotations. ChartY-en and ChartY-zh are crucial for evaluating the model's adaptability and robustness across different languages, rendering technologies and aesthetics.

By redefining the ground truth formats in Python-dict and incorporating a wider variety of chart images, our benchmark aims to provide a more comprehensive and rigorous evaluation of models' capabilities in extracting structured information.

**Table 3: Comparison of raw and purified prediction AP scores across datasets for Structural Extraction (SE) task. Utilizing a self-consistency distance threshold of 0.1.**

| | ChartQA-SE | | PlotQA-SE | | ChartX-SE | | ChartY-en | | ChartY-zh | |
|---|---|---|---|---|---|---|---|---|---|---|
| | Raw | Purified | Raw | Purified | Raw | Purified | Raw | Purified | Raw | Purified |
| Image Samples | 1509 | 1174 | 33657 | 23723 | 2360 | 1429 | 4000 | 3026 | 1991 | 1662 |
| AP@strict ■ | 72.02 | 81.97 (+9.95) | 34.56 | 36.58 (+2.02) | 44.55 | 53.32 (+8.77) | 70.06 | 78.51 (+8.45) | 76.14 | 83.56 (+7.42) |
| AP@slight ■ | 82.91 | 92.46 (+9.55) | 84.18 | 88.81 (+4.63) | 53.12 | 61.94 (+8.82) | 82.12 | 89.00 (+6.88) | 85.64 | 91.79 (+6.15) |
| AP@high ■ | 85.94 | 94.86 (+8.92) | 86.10 | 90.72 (+4.62) | 59.72 | 68.64 (+8.92) | 86.15 | 92.19 (+6.04) | 89.37 | 94.73 (+5.36) |

**Table 4: Ablation of auxiliary token's position. ChartQA-SE and PlotQA-SE scores are reported. AP ■, AP ■ and AP ■ represent AP@strict, AP@slight, and AP@high, respectively.**

| | Front | Behind | AP ■ | AP ■ | AP ■ |
|---|---|---|---|---|---|
| ChartQA-SE | – | – | 70.95 | 82.25 | 85.06 |
| | ✓ | | 72.02 | 82.91 | 85.94 |
| | | ✓ | 68.63 | 80.11 | 83.60 |
| PlotQA-SE | – | – | 30.50 | 79.65 | 81.78 |
| | ✓ | | 34.56 | 84.18 | 86.10 |
| | | ✓ | 26.59 | 75.12 | 77.68 |

## 4.3 Comparison with State-of-the-Arts

As indicated in Table 2, our model, OneChart, consistently achieve excellent AP for SE tasks across multiple chart sources and types, while having the smallest size (0.2B). Specifically, for datasets like ChartQA-SE and the "barnum", "linenum" and "pie" in ChartX-SE, which have numerical values directly marked on chart images, with the data-driven enhancement for chart vision, OneChart showcases pleasing performance. When it comes to parsing chart images without clear numerical markers, where the model needs to derive values by aligning with the coordinate axes (as primarily seen in PlotQA-SE and the "bar" and "line" in ChartX-SE), OneChart performs AP@strict above the suboptimal methods by 19.1% to 29.38%. It is worth noting that this improvement significantly surpasses the model's performance in tasks involving charts without numerical markers. This showcases OneChart's excellent perceptual alignment ability and the precise ability to derive numbers, bolstered by the use of an auxiliary token. Moreover, in both ChartY-en and ChartY-zh, OneChart's performance far exceeds other models, indicating robustness across different styles and languages.

In the textual OCR task, OneChart achieves an average OCR score exceeding 90 across all datasets, indicating its capability to deliver clear chart meanings, which provides a solid foundation for downstream QA tasks.

## 4.4 Ablation Studies

We initially conduct two ablation studies on the proposed auxiliary token. The specific experimental results are shown in Table 3 and Table 4.

The presence or absence of the auxiliary token and the impact of its placement on model performance are recorded in Table 4. It

**Table 5: Ablation of training strategies. ChartQA-SE scores are reported.**

| Stage2 | Stage3 | AP@strict | AP@slight | AP@high |
|---|---|---|---|---|
| ✓ | | 68.93 | 79.64 | 82.66 |
| | ✓ | 68.42 | 80.21 | 83.52 |
| ✓ | ✓ | 72.02 | 82.91 | 85.94 |

can be seen from it that the placement at the beginning of the sequence is beneficial, yielding higher AP scores across all evaluation tolerances. This can be attributed to the model's ability to leverage causal attention mechanisms to immediately attend to the initial embeddings, directly influencing the text output which dictates the prediction results. Conversely, when the token is placed at the end, the text output cannot effectively utilize these embeddings due to the causal attention's unidirectional nature. Moreover, the presence of a number loss at the end might introduce noise, further impeding the textual learning process. Therefore, we believe that the introduction of the auxiliary token effectively improves the performance of the model, provided that it is placed in a reasonable position.

In addition to enhancing the model's parsing ability for chart images, as discussed in Section 3.5, the introduction of the auxiliary token and the design of the corresponding decoder also enable the model to perform the reliability check on its own output. To demonstrate the effectiveness of our designed reliability check method during the inference process, we filter model's original outputs by setting a self-consistency distance threshold of $\delta = 0.1$ (as detailed in Section 3.5) to obtain purified outputs, and calculate their AP scores alongside original outputs. The results are displayed in Table 3. The "Raw" and "Purified" categories represent the original and filtered outputs, respectively. Notably, after removing "unreliable" results identified through reliability checks, the purified outputs in the ChartQA-SE benchmark show an impressive 9.95% increase in AP@strict compared to the original results. In the other four benchmarks, the purified results show an increase in AP@strict ranging from 2.02% to 8.77%. This highlights that the introduced auxiliary token endows our model with the inherent ability to effectively evaluate its output accuracy, which is quite remarkable.

Our model undergoes a three-stage training process. In Stage2, training is confined to the language model and the auxiliary decoder, while in Stage3, the entire model undergoes training. We conduct a thorough analysis of various training methodologies, with the

**Table 6: Fully-supervised and Zero/One-shot results on ChartQA benchmark. Fig. represents with or without chart figure input. mPLUG. represents mPLUG-DocOwl model.**

| | Method | Fig. | ChartQA | | |
|---|---|---|---|---|---|
| | | | aug. | human | avg. |
| Fully-supervised | LLaVA1.5 [11] | ✓ | 13.4 | 21.6 | 17.5 |
| | LLaVA1.6 [24] | ✓ | 66.1 | 46.0 | 56.0 |
| | Pix2Struct [41] | ✓ | 81.6 | 30.5 | 56.0 |
| | mPLUG. [42] | ✓ | - | - | 57.4 |
| | Vary-toy [15] | ✓ | 84.8 | 33.4 | 59.1 |
| | QwenVL [43] | ✓ | 83.6 | 41.6 | 62.6 |
| | Matcha [19] | ✓ | 90.2 | 38.2 | 64.2 |
| Zero/One-shot | DePlot+GPT3.5 [28] | ✗ | 37.3 | 36.5 | 36.9 |
| | Ours+GPT3.5 | ✗ | 73.0 | 42.0 | 57.5 |
| | Ours+LLaVA1.5 | ✗ | 63.4 | 24.4 | 43.9 |
| | | | (+50) | (+2.8) | (+26.4) |
| | Ours+LLaVA1.5 | ✓ | 69.4 | 30.9 | 50.1 |
| | | | (+56) | (+9.3) | (+32.6) |
| | Ours+LLaVA1.6 | ✓ | 85.3 | 49.1 | 67.2 |
| | | | (+19.2) | (+3.1) | (+11.2) |

outcomes detailed in Table 5. For fair comparison, when the model is only trained on Stage2 or Stage3, the model is trained on 2 epochs. When the model is trained on Stage2 and Stage3, train one epoch on each stage. We observe that training auxiliary decoder solely on the Stage2 or Stage3 is inadequate. Optimal results are achieved by initially warming up the auxiliary decoder, followed by a fine-tuning of the entire model for 1 epoch.

## 4.5 QA Performance

In order to further analyze and demonstrate the effectiveness of the proposed OneChart, it is combined with large language models (LLMs) and vision-language models (VLMs), and compared with existing methods on downstream tasks.

Table 6 provides an overview of the comparative analysis of QA task accuracy on the ChartQA dataset. A prominent aspect of our approach is the use of one shot methods to enable GPT-3.5, which lacks visual functionality, to answer chart related questions. Compared with the results obtained from CSV generated using GPT-3.5 and DePlot, our model exhibits significantly better performance (36.9%→57.5%).

In addition, the QA capability of VLMs with inherent visual understanding, such as LLaVA [11], has been enhanced solely through zero shot using the dictionary we have analyzed. When these VLMs are equipped with both Chart images and our structured parsing dict, their overall QA has improved (17.5%→50.1% for LLaVA1.5, 56.0%→67.2% for LLaVA1.6). The results highlight the versatility and effectiveness of our method in promoting more accurate chart understanding and information retrieval between different artificial intelligence systems.

## 5 CONCLUSION

In this study, we introduce OneChart, an innovative framework designed to revolutionize the process of interpreting and extracting information from charts and plots. By leveraging an end-to-end autoregressive method, this approach transforms chart images into Python-dict formatted text, enhancing both efficiency and accuracy. OneChart is reinforced by the introduction of a auxiliary special token and the integration of a custom L1 loss alongside the language cross-entropy loss. This methodology not only minimizes ambiguity in language supervision and improves the accuracy of structural extraction, but also introduces a reliable scoring system to purify the output during inference.

Additionally, by establishing the ChartY benchmark, we provide a comprehensive evaluation tool for chart comprehension, addressing the inadequacies of existing QA-type benchmarks. Overall, OneChart represents a substantial advancement in the field, demonstrating an average 20% improvement in information extraction from a variety of chart types, while maintaining minimal model size. Additionally, the positive outcomes of this study highlight the importance of specialized loss functions in model training for specific tasks. In the future, we will focus on expanding OneChart's capabilities to include more diverse and complex chart types and exploring its application in real-world scenarios.

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
