# OpenReview forum: "OneChart: Purify the Chart Structural Extraction via One Auxiliary Token"
_acmmm.org/ACMMM/2024/Conference — MM2024 Oral_

### Official Review · Reviewer_yYPy · 2024-05-24

**Rating:** 4
**Confidence:** 3

**Summary:**

In this paper, the authors propose a char-to-dict model named OneChart. The OneChart model employs an encoder-decoder architecture, using SAM-base as the vision encoder and OPT-125M as the decoder.

The latest chart understanding models are fundamentally based on VLMs, but the authors point out that existing VLMs have two key shortcomings: CLIP bias and insensitivity to numerical values. To address the issue of CLIP bias, the authors re-pretrained OneChart. They utilized large models to generate a substantial amount of JSON data representing charts and leveraged `pyecharts` and `matplotlib` to produce images of the charts. To tackle the insensitivity to numerical values, the authors introduced an auxiliary token and a decoder, requiring the decoder to generate the numerical values in the chart based on the embedding of the auxiliary token.

**Strengths:**

1. The introduction of the auxiliary token and the decoder is an novel approach that effectively utilizes the characteristics of transformers.

2. Compared to existing state-of-the-art (SOTA) models, the OneChart model has achieved a significant lead in performance.

3. The paper introduces a new benchmark called ChartY, which can be helpful to the research field.

**Limitations:**

1. The authors believe that VLMs have a CLIP bias, which affects their performance in chart understanding tasks, but they have not proved the existence of CLIP bias through experiments or case studies.

2. I am concerned whether the OneChart model will overfit to the layout of charts generated by pyecharts and matplotlib.

3. The authors did not compare OneChart with the latest multimodal LLM APIs in their experiments.

**Suitability:**

3

---

### Official Review · Reviewer_bdc6 · 2024-05-25

**Rating:** 5
**Confidence:** 3

**Summary:**

This paper focus on the task of chart parsing. Specifically, it attempts to convert charts into Python-dict format. To this end, this paper first constructs a large scale datasets based on the gpt3.5. Then, it pretrains the opt-125M using the dataset. In addition, this paper designs an auxiliary number decoder and proposes an auxiliary number decoder to enhance the capacity of the model. Results on several benchmarks show the advantages of the proposed model.

**Strengths:**

1. This paper attempts an effective data synthesis method.
2. The proposed method outperforms state-of-the-art chart parsing models on several benchmarks, which has proved its effectiveness.

**Limitations:**

1. The description of the data synthesis method is not enough.
2. This method is not verified on a larger model.

**Suitability:**

3

---

### Official Review · Reviewer_jfXf · 2024-05-28

**Rating:** 4
**Confidence:** 4

**Summary:**

This paper proposes an agent specifically devised for the structural extraction of chart information (OneChart), which introduces an auxiliary token and an additional decoder to improve the accuracy of numerical value parsing. Meanwhile, it presents a new benchmark ChartY spanning a wide array of topics, chart types, and languages, offering a robust platform for future research and evaluation. Experiments demonstrate that OneChart achieves SOTA performance in structural extraction and the proposed auxiliary token's effectiveness.

**Strengths:**

1. The paper proposes a OneChart model designing an auxiliary token to achieve more accurate numerical value parsing.
2. The paper creates a new ChartY benchmark spanning a wide array of topics, chart types, and languages, offering a robust platform for future research and evaluation.
3. The paper conducts comprehensive experiments and analyses to evaluate the proposed OneChart’s chart structural extraction capability and the effectiveness of the auxiliary token.

**Limitations:**

1. OneChart mainly focuses on Chart Structural Extraction and Factoid Chart Question Answering tasks, it would be beneficial to include more diverse chart-related tasks (e.g., Complex Question Answering on OpenCQA [1], Chart Summarization on Chart-to-Text [2]) to further demonstrate the effectiveness of OneChart. As mentioned in Line 143-144, "This model serves as a foundational framework for other researchers to further develop and enhance.", I wonder whether OneChart have chart reasoning capability to further solve practical issues.
2. Need to add more lastest baselines for evaluation, such as UniChart [3] and MMC [4].
3. OneChart undergoes a three-stage training process with a number of training data and time (mentioned in Section 3.4), can you discuss the efficiency of OneChart during inference, including inference time?
4. It is helpful for us to release your code to further comprehend the proposed OneChart.

**Reference**
[1] Opencqa: Open-ended question answering with charts, EMNLP, 2022.
[2] Chart-to-text: A large-scale benchmark for chart summarization, ACL, 2022.
[3] UniChart: A Universal Vision-language Pretrained Model for Chart Comprehension and Reasoning, EMNLP, 2023.
[4] MMC: Advancing Multimodal Chart Understanding with Large-scale Instruction Tuning, NAACL, 2024.

**Suitability:**

3

---

### Official Review · Reviewer_p2at · 2024-05-29

**Rating:** 4
**Confidence:** 4

**Summary:**

This work focuses on Chart parsing. To address a significant challenge due to the diversity of styles, values, and texts, they propose
OneChart: a reliable agent specifically devised for the structural extraction of chart information. Specifically, to enhance the
reliability of the numerical parts of the output, they introduce an auxiliary token placed at the beginning of the total tokens along with
an additional decoder. The numerically optimized (auxiliary) token allows subsequent tokens for chart parsing to capture enhanced
numerical features through causal attention. Experiments show the effectiveness of the proposed method.

**Strengths:**

1) better results with an auxiliary token, i.e., the proposed method
2) offer a comprehensive evaluation tool for chart comprehension

**Limitations:**

1) the method is effective, yet so simple

**Suitability:**

2

---

### Official Review · Reviewer_5YU6 · 2024-05-29

**Rating:** 5
**Confidence:** 3

**Summary:**

The authors propose OneChart, which is an end-to-end vision-language model that incorporates an auxiliary token and a custom decoder to enhance the accuracy of numerical value parsing in charts. The authors introduce a new benchmark, ChartY, comprising diverse chart types and languages to evaluate the model's performance.

**Strengths:**

1. The introduction of an auxiliary token and a specialized decoder for numerical value parsing is an innovative idea that addresses a critical challenge in chart understanding.
2. The ChartY benchmark covers a wide range of chart types, topics, and languages, providing a more robust evaluation platform for chart parsing models.
3. The authors conduct extensive experiments and ablation studies to validate the effectiveness of their approach, showcasing significant improvements over state-of-the-art methods, especially for charts without numerical annotations.
4. The authors demonstrate the utility of OneChart in improving the performance of large language models and vision-language models on downstream chart question answering tasks.

**Limitations:**

1. While the authors claim that OneChart can handle diverse chart types, the experiments primarily focus on bar, line, and pie charts. The model's performance on more complex chart types, such as scatter plots, area charts, or composite charts, is not extensively evaluated.
2. The paper primarily reports quantitative metrics but lacks a qualitative analysis of the model's output, which could provide insights into the strengths and weaknesses of the proposed approach.

**Suitability:**

3

---

### Meta-Review · Area_Chair_f4rj · 2024-06-27

**Recommendation:** Accept (Oral)
**Confidence:** 5

**Metareview:**

This paper presents a good approach to improving numerical value parsing in charts through the introduction of an auxiliary token and a custom decoder. Results have shown that it has enhanced the performance of recent VLMs. The authors provide a comprehensive evaluation using the newly introduced ChartY benchmark, which covers a diverse range of chart types and languages. Extensive experiments and ablation studies validate OneChart's effectiveness, showing improvements over state-of-the-art methods, particularly for charts without numerical annotations. Despite some limitations, such as the focus on a limited range of chart types (primarily bar, line, and pie charts), the absence of qualitative analysis, and the need for evaluation on more diverse chart-related tasks, the paper still can be of good interest to ACM MM community. The reviewers generally leaned towards acceptance, acknowledging the potential impact and significance of the work. Considering the strong arguments presented in the rebuttal and the updates from the reviewers, particularly the change from weak accept to accept by Reviewer 5YU6, it makes paper a strong candidate for the conference. Therefore, I recommend that this submission be accepted and presented at ACM Multimedia 2024, with the expectation that the authors will address the noted limitations in future research.